# The Metabolic Benefits of Menopausal Hormone Therapy Are Not Mediated by Improved Nutritional Habits. The OsteoLaus Cohort

**DOI:** 10.3390/nu11081930

**Published:** 2019-08-16

**Authors:** Georgios E. Papadakis, Didier Hans, Elena Gonzalez Rodriguez, Peter Vollenweider, Gerard Waeber, Pedro Marques-Vidal, Olivier Lamy

**Affiliations:** 1Service of Endocrinology, Diabetes and Metabolism, Lausanne University Hospital and University of Lausanne, CH-1011 Lausanne, Switzerland; 2Center of Bone Diseases, CHUV, Lausanne University Hospital and University of Lausanne, CH-1011 Lausanne, Switzerland; 3Service of Internal Medicine, CHUV, Lausanne University Hospital and University of Lausanne, CH-1011 Lausanne, Switzerland

**Keywords:** menopause, estrogens, estrogen deficiency, visceral fat, menopausal hormone therapy, energy intake, macronutrients, dietary patterns, dietary recommendations

## Abstract

Menopause alters body composition by increasing fat mass. Menopausal hormone therapy (MHT) is associated with decreased total and visceral adiposity. It is unclear whether MHT favorably affects energy intake. We aimed to assess in the OsteoLaus cohort whether total energy intake (TEI) and/or diet quality (macro- and micronutrients, dietary patterns, dietary scores, dietary recommendations)—evaluated by a validated food frequency questionnaire—differ in 839 postmenopausal women classified as current, past or never MHT users. There was no difference between groups regarding TEI or consumption of macronutrients. After multivariable adjustment, MHT users were less likely to adhere to the unhealthy pattern ‘fat and sugar: Current vs. never users [OR (95% CI): 0.48 (0.28–0.82)]; past vs. never users [OR (95% CI): 0.47 (0.27–0.78)]. Past users exhibited a better performance in the revised score for Mediterranean diet than never users (5.00 ± 0.12 vs. 4.63 ± 0.08, *p* < 0.04). Differences regarding compliance with dietary recommendations were no longer significant after adjustment for covariates. Overall, these results argue against a major role of TEI and diet quality as possible mediators of the MHT metabolic benefits. Future research on this relationship should focus on other potential targets of MHT, such as resting energy expenditure and physical activity.

## 1. Introduction

Among the most pronounced expressions of sexual dimorphism, menopause or more accurately the menopausal transition (MT) is a phase of progressive gonadal insufficiency occurring selectively in women and evolving towards the permanent cessation of menses at a mean age of 50–52 years [1]. Estrogen deficiency is the principal underlying hormonal consequence. In addition to the well-known vasomotor symptoms [2] and accelerated bone loss [3], MT significantly alters body composition with opposite effects on fat and lean body mass [4]. The Study of Women’s Health Across the Nation cohort, which followed longitudinally 1246 women of different ethnic backgrounds, confirmed that both the accelerated fat mass (FM) gain and the decline in lean mass are menopause-related and not merely age-dependent [5]. Several small interventional studies evaluated whether these changes are reversible with menopausal hormone therapy (MHT) and by what mechanisms [6]. These studies have yielded mixed results.

Recently, we analyzed the postmenopausal women of OsteoLaus cohort and demonstrated that MHT prevents the increase in fat mass, and, in particular, the increase of visceral adipose tissue (VAT) [6]. It is still unclear whether these benefits result from the direct effect of estrogens on fat tissue and muscles [7,8], or whether changes in behavior influence the balance between energy intake and expenditure [9]. In favor of the latter hypothesis, evidence suggests that female reproductive hormones modulate the physiologic control of food intake [10,11]. Caloric intake progressively declines during the follicular phase following an increase of serum estradiol (E2) [12]. Loss of estrogens following ovariectomy elicits overeating in animal models, including monkeys and rats [13], a finding partly reversed by E2 administration [12,14]. Studies in transgenic mice highlighted the activation of estrogen receptor type-α in the hypothalamic pro-opiomelanocortin neurons of the arcuate nucleus as the principal mechanism underlying the anorectic effects of E2 [15]. There is, however, a dearth of research exploring whether exogenous estrogens, and in particular MHT, affect the neuroendocrine control of eating in humans as well.

An initial evaluation of the OsteoLaus cohort showed no difference in total caloric intake between postmenopausal women on MHT and those who were MHT-naïve [6]. Given the promising preclinical data mentioned above, we hypothesized that improving the quality of nutrition can contribute to the reduction of FM and VAT induced by MHT. The objective of this study was to explore whether relative macronutrient intake, adherence to healthy eating habits and other qualitative elements contribute to the body composition advantages in postmenopausal women of the OsteoLaus cohort taking MHT.

## 2. Material and Methods

### 2.1. Setting

OsteoLaus is a sub-study of the CoLaus/PsycoLaus study, an ongoing prospective study aiming to assess the determinants of cardiovascular disease using a population-based sample drawn from the city of Lausanne, Switzerland [16]. Between September 2009 and September 2012, all women aged between 50 and 80 years from the CoLaus/PsycoLaus study were invited to participate in the OsteoLaus sub-study and approximately 85% accepted. The primary aims of OsteoLaus are to compare different models of fracture risk prediction and to assess the relationship between osteoporosis and cardiovascular diseases [17]. CoLaus and OsteoLaus studies were approved by the Institutional Ethics Committee of the University of Lausanne. All participants signed an informed consent form.

### 2.2. Participants

Of the 1475 postmenopausal women included in OsteoLaus, 1053 had a body composition assessed by Dual X-ray absorptiometry (Discovery A System, Hologic, Inc., Marlborough, MA, USA). Women treated with estrogen-mediated effects (aromatase inhibitors, tamoxifen, antiandrogens) were excluded from this analysis [18]. The participants were subsequently divided into three groups: Current (CU), past (PU) and never users (NU) of MHT. CU were taking MHT at trial entry or discontinued treatment less than six months before trial entry. PU followed MHT for at least six months (otherwise considered as NU) and discontinued MHT at least six months before trial entry (otherwise considered as CU) [18].

### 2.3. Dietary Data

Dietary data for this group were derived from the nearest CoLaus visit (second visit) who took place within six months before the OsteoLaus inclusion. Dietary intake was assessed using a self-administered, semi-quantitative Food Frequency Questionnaire (FFQ), which also includes portion size and has been validated against 24 h recalls among 626 volunteers from the Geneva population [19]. Data derived from this FFQ have contributed to local and worldwide analysis [20,21]. Concisely, this FFQ assesses the dietary intake of the previous four weeks and consists of 97 different food items that account for more than 90% of the intake of calories, proteins, fat, carbohydrates, alcohol, cholesterol, vitamin D and retinol, and 85% of fiber, carotene and iron. Conversion of the FFQ responses into nutrients was based on the French CIQUAL food composition table. Total energy intake (TEI) was computed, including alcohol consumption.

Dietary patterns were derived using principal components analysis (PCA) based on food consumption frequencies. Food consumption frequencies were defined as follows: Never during the past four weeks = 0; 1 time per month = 1/28; 2–3 times per month = 2.5/28; 1–2 times per week = 1.5/7; 3–4 times per week = 3.5/7; once per day = 1, and ≥ twice per day = 2.5. We identified three dietary patterns: “Meat and fries”, “fruits and vegetables” and “fatty and sugary”. A detailed description of the assessment and characteristics of the dietary patterns is provided elsewhere [22].

Three health diet scores were additionally computed: Two Mediterranean diet scores (the classic [23] and a more recent one adapted to Switzerland [24], as well as the Alternate Healthy Eating Index (AHEI) developed by the Harvard School of Public Health [25]. The particularities of each score, as well as their validation in a Swiss population, have been previously reported [26]. Dietary patterns and diet scores were further categorized into quintiles, and the prevalence of participants in the highest quartile, according to MHT group was assessed.

Participants were dichotomized according to whether they followed the dietary recommendations for fruits, vegetables, meat, fish and dairy products from the Swiss Society of Nutrition (Schweizerische Gesellschaft für Ernährung SGE, 2013). The recommendations were ≥2 fruit portions/day; ≥3 vegetable portions/day; ≤5 meat portions/week; ≥1 fish portion/week and ≥3 dairy products portions/day [27]. As the FFQ queried about fresh and fried fish, two categories were considered: One included and one excluded fried fish. Participants were further dichotomized if they complied with at least three recommendations or not; two categories of compliance with at least three recommendations were created, depending on the type of fish consumed (all or fresh only).

### 2.4. Covariates

Bodyweight and height were measured using standard procedures [16], and body mass index (BMI) was defined as weight (kg)/height (m)^2^. Overweight was defined as 25 ≤ BMI < 30 kg/m^2^ and obesity as BMI ≥ 30 kg/m^2^. Smoking status was defined as never, former (irrespective of the time since quitting) and current (irrespective of the amount smoked). Diabetes status was defined as a fasting plasma glucose ≥7.0 mmol/L and/or presence of insulin or oral antidiabetic drugs. Physical activity was estimated by a self-administered physical activity frequency questionnaire (PAFQ). The questionnaire listed 70 activities or groups of activities and was validated against the measurement of energy expenditure by heart rate monitor with satisfactory correlations (r = 0.76) between the two methods [28]. For this analysis, only sedentary status (yes/no) was used. Sedentary status was defined when the participant spent less than 10% of her total daily energy expenditure in activities with an intensity over four basal metabolic rate equivalents. Trained collaborators performed the examinations, interviewed the participants, and checked the self-administered questionnaires for completion.

### 2.5. Statistical Analysis

Statistical analyses were conducted using Stata v15.1 (StataCorp, College Station, TX, USA) for Windows. Descriptive results were expressed as a number of participants (percentage) for categorical variables or as average ± standard deviation for continuous variables. Bivariate analyses were conducted using chi-square for categorical variables and Kruskal-Wallis test for continuous variables. Multivariable analyses (adjusting for total energy intake, age, education, BMI category, sedentary level and diabetes) were conducted using analysis of variance and results were expressed as adjusted average ± standard error. Statistical significance was considered for a two-tailed test with a *p*-value < 0.05.

## 3. Results

### 3.1. Selection of Participants and Characteristics of the Final Sample

Out of the initial sample of 1053 women, 214 (20.3%) were excluded mainly due to absent or incomplete dietary data. The detailed reasons for exclusion are illustrated in Figure 1. The characteristics of included and excluded participants are summarized in Appendix A. Excluded subjects were significantly older, less educated and more frequently obese and diabetic. The characteristics of the included participants according to MHT status, are shown in Table 1. The three groups differed in age (PU > CU > NU, *p* < 0.001). CU were significantly thinner than NU (−0.9 kg/m^2^), which was also reflected by means of a lower percentage of obesity in this group. There was no significant difference in terms of educational level or current/past smoking. CU tended to be less sedentary, and be associated with a lower incidence of diabetes, though without reaching statistical significance.

### 3.2. Menopausal Hormone Therapy and Dietary Intake

Data on the consumption of different nutrients according to MHT status are displayed in Table 2. The three groups did not significantly differ regarding TEI. Analysis of macronutrients showed similar results between groups except for higher vegetal protein and fiber intake in PU, which did not persist after multivariate adjustment. In particular, CU did not exhibit a lower consumption of carbohydrates or saturated fatty acids (SFA) compared with NU. PU significantly exceeded the other groups in terms of iron intake, as well as consumption of fruits when combined with juices.

### 3.3. Menopause Hormone Therapy and Dietary Patterns

Three patterns were examined, as recently identified in the French-speaking population of Switzerland [20]: Meat and chips, fruits and vegetables, and fat and sugar. Adherence to each dietary pattern according to MHT status is shown in Table 2 (negative scores indicating low adherence and vice versa), whereas the prevalence rate ratios (PRRs, and 95% confidence interval) of being in the highest quintile relative to the other four are illustrated in Table 3. Overall, all groups were associated with low adherence to the meat and chips. A significantly higher adherence for the fruits and vegetables was observed in PU, a finding that was lessened following multivariate adjustments. Both MHT groups exhibited a tendency to negative scores for the fat and sugar pattern in contrast to NU. When focusing the analysis to the probability of being in the highest quintile for this pattern (Table 3), both CU and PU had significantly lower PRRs in comparison to NU.

### 3.4. Menopause Hormone Therapy and Dietary Scores

Mean values of dietary scores according to MHT status, are summarized in Table 2. Both CU and PU groups scored higher than NU for all three respective scores with their performance becoming statistically higher for the revised Mediterranean dietary score. However, this difference did not translate to a significantly higher odds ratio for being in the highest quintile for any of the examined scores (Table 3).

### 3.5. Menopause Hormone Therapy and Dietary Recommendations

Compliance with dietary recommendations according to MHT status is illustrated in Table 3. The bivariate analysis was remarkable for higher adherence of PU and to a lesser extent CU to the intake of at least two fruits per day and the respect of at least three recommendations overall. Nevertheless, these differences were no longer statistically significant after multivariate adjustment.

## 4. Discussion

In this cross-sectional analysis of the OsteoLaus cohort, no significant differences regarding dietary intake were found according to MHT status. These results argue against a major role of caloric intake and diet quality as possible mediators of the metabolic benefits of MHT.

### 4.1. Menopause Hormone Therapy, Weight Change and Caloric Intake

Though less extensively studied than the impact of MT on bone, a trend towards weight gain and visceral fat accumulation is regularly found in longitudinal studies [5,29]. In the current study, women taking MHT exhibited lower BMI, which was not explained by reduced TEI. These results are in agreement with the few available human studies. Lovejoy et al. followed longitudinally 156 middle-aged women for four years, during which 51 became postmenopausal [29]. Postmenopausal women selectively gained more VAT, a finding accompanied by decreased sleeping energy expenditure and fat oxidation. No increase in caloric intake was observed during the MT. A large observational study of healthy perimenopausal women (*n* = 907) detected postmenopausal weight gain independently of MHT use, and without a concomitant increase of food intake [30]. The only available randomized, double-blind, placebo-controlled trial that assessed energy intake was conducted in a small sample size (*n* = 14 per group) and showed a non-significant 12% decrease in the MHT group over two years [31]. Caloric intake did not change after one year of MHT in two other small non-randomized studies [32,33].

In our study, a detailed analysis of macro- and micronutrient consumption did not reveal any beneficial differences in CU, notably regarding SFA, which are traditionally considered as metabolically deleterious. A link between estrogen and food composition is suggested by an animal study in which oophorectomized rats exhibited hyperphagia with a predominant increase in dietary fat intake secondary to estrogen deficiency [13]. In their longitudinal study across the MT, Lovejoy et al. observed increased cholesterol and SFA intake during the first post-menopausal years [29]. Only a few non-randomized and small-sized studies have assessed the effect of MHT on food composition and did not detect any preferential intake of macronutrients [32,33,34]. It is possible that MHT does not fully reverse the macronutrient preferences induced by menopause, due to significant differences in comparison with pre-menopausal sex steroids levels. Notably, the majority of MHT regimes implement continuous progestin doses, which is in contrast to the cyclic rise of progesterone in naturally menstruating women. Progestins have been shown to antagonize estrogens and promote binge eating, which may explain in part the observed differences [35].

### 4.2. Menopause Hormone Therapy and Diet Quality (Dietary Patterns, Scores, Recommendations)

The effect of estrogens on energy intake may not be limited to quantitative changes, but also encompass changes in diet quality. In a large Australia study, the eating habits of more than 1500 participants aged 25 to 75 were monitored for more than 15 years [36]. The results highlighted that MHT is an independent factor associated with improved diet quality in women. Besides the analysis of diet composition, diet quality can be assessed by other parameters, such as dietary patterns, dietary scores, and compliance with dietary recommendations and. To our knowledge, this is the first study evaluating the effect of MHT on such parameters.

Dietary patterns are a reliable indicator for an overall assessment of individual’s diet [37] and have been validated to be strongly associated with chronic diseases such as type 2 diabetes [38]. Overall, CU had a tendency to a lower adherence to both unhealthy patterns and a statistically significant higher adherence to ‘fruits and vegetables’ pattern as compared to NU. But these differences disappeared after adjustment for covariates. However, CU were significantly less likely to be in the highest quintile for the ‘fat and sugar’ pattern than NU, even after multivariate analysis. Interestingly, this pattern exhibited strong correlations with total energy intake and saturated fat among CoLaus participants [22]. However, in the absence of other concordant results on macronutrients analysis, and given the cross-sectional design of the study, this isolated finding should be interpreted with caution. In addition, this pattern was also present in PU which did not show any body composition advantage in our previous study [6].

Interestingly, a statistical difference (PU > CU > NU) was detected for the revised Mediterranean score, while no difference was found for the other ones. The revised Mediterranean score is considered to be more adapted to the Swiss population [24]. PU scored even higher than CU, an advantage deriving from a preferential consumption of fruits and vegetables in this group.

In agreement with the majority of our results and following a multivariate adjustment, the three groups did not differ regarding compliance with the Swiss dietary recommendations. There are no previous reports regarding the effect of MHT or other estrogen-based preparations on this outcome. Recently, an analysis of the PsyCoLaus cohort, which includes all the participants of the OsteoLaus cohort, did not find any significant association between adherence to the same dietary recommendations and the incidence of major depressive disorder [39].

### 4.3. Study Limitations and Strengths

This study has several limitations. First, excluded participants differed significantly from those included in the analysis. Hence, it is possible that our findings do not apply to the whole population of postmenopausal women. The cross-sectional setting of the study only allows establishing associations, and no causal inferences can be drawn. As with all observation trials on MHT, it is possible that some of the differences are due to a selection bias, given that women starting MHT tend to have a healthier lifestyle [40]. However, there was no difference between groups in terms of education level and smoking, two cardinal components of the ‘healthy women bias’ in previous studies. Information regarding the type of MHT (estrogen-alone or estrogen/progestin) and the route of administration (oral, transdermal, vaginal) was self-reported, preventing us from reliably assessing these factors and their differential effect on body composition or nutrition. Further, we were unable to verify the adherence of participants to MHT. The vast majority of participants were Caucasians, limiting the generalization of the study’s conclusions to women of different ethnic backgrounds.

Conversely, our study has several strengths. To the best of our knowledge, this is the first large-scale transversal study that thoroughly explores both quantitative and qualitative aspects of dietary behavior according to MHT status. The implementation of several parameters (dietary patterns, compliance with dietary guidelines, macro- and micronutrients estimations) allowed for a global approach with different tools complementing each other. In addition, current and past MHT users were distinguished. The large sample of the OsteoLaus cohort allowed for adequate statistical power. All nutritional assessment was based on standardized tools which had been previously tested and validated in the French-speaking population of Switzerland.

## 5. Conclusions

In conclusion, we did not observe any meaningful associations between MHT users and improved eating habits. It is, therefore, unlikely that improved dietary behavior could explain the previously observed reduction of total and visceral adiposity in current MHT users. Future research on the relationship between estrogens and body composition should focus on other potential modifiers, such as resting energy expenditure and physical activity.

## Figures and Tables

**Figure 1 nutrients-11-01930-f001:**
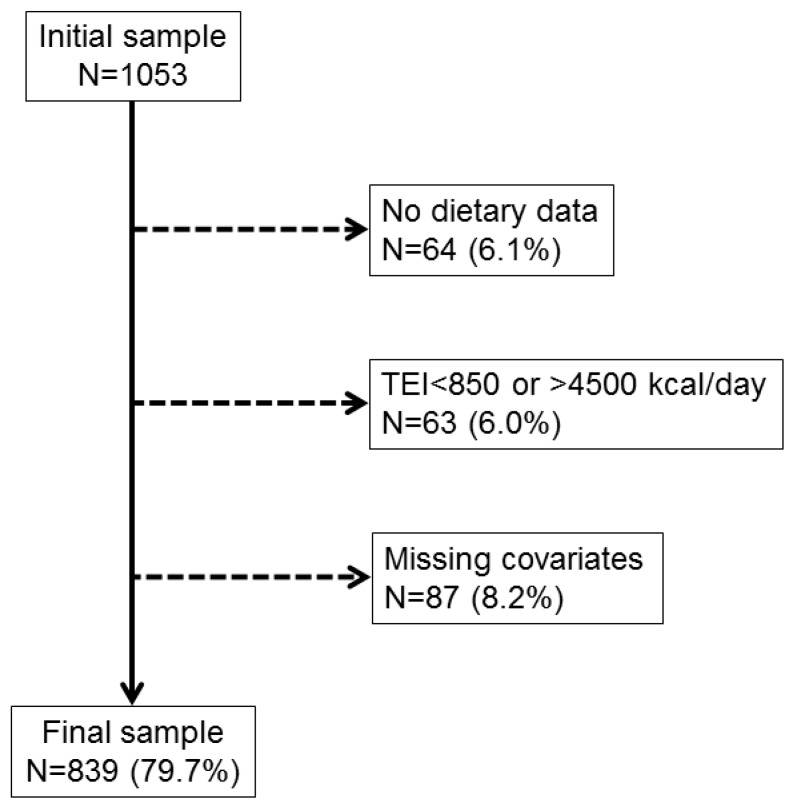
Flow chart of the study. Selection procedure of the participants of the OsteoLaus study, Lausanne, Switzerland. TEI, Total Energy Intake.

**Table 1 nutrients-11-01930-t001:** Characteristics of the participants according to menopausal hormone therapy.

Characteristic	Never	Current	Past	*p*-Value
Number	432	180	227	
Age (years)	60.8 ± 7.5	62.5 ± 6.8	66.9 ± 6.2	**<0.001**
Educational level (%)				0.177
University	71 (16.4)	34 (18.9)	36 (15.9)	
High school	115 (26.6)	47 (26.1)	58 (25.6)	
Apprenticeship	174 (40.3)	77 (42.8)	81 (35.7)	
Mandatory	72 (16.7)	22 (12.2)	52 (22.9)	
Smoking categories (%)				0.413
Never	178 (41.2)	81 (45.0)	108 (47.6)	
Former	167 (38.7)	69 (38.3)	85 (37.4)	
Current	87 (20.1)	30 (16.7)	34 (15.0)	
Body mass index (kg/m^2^)	25.6 ± 4.1	24.7 ± 3.9	25.4 ± 4.1	**0.032**
BMI categories (%)				**0.047**
Normal	207 (47.9)	101 (56.1)	106 (46.7)	
Overweight	151 (35.0)	63 (35.0)	91 (40.1)	
Obese	74 (17.1)	16 (8.9)	30 (13.2)	
Sedentary status (%)	279 (64.6)	109 (60.6)	155 (68.3)	0.268
Diabetes (%)	24 (5.6)	5 (2.8)	11 (4.9)	0.334

BMI, body mass index. Results are expressed as a number of participants (percentage) for categorical variables and as average ± standard deviation for continuous variables. Between-group comparisons performed using chi-square for categorical variables and analysis of variance for continuous variables. Statistically significant p-values are highlighted in bold.

**Table 2 nutrients-11-01930-t002:** Dietary intake according to menopausal hormone therapy.

	Bivariate Analysis				Multivariable Analysis			
	Never	Current	Past	*p*-Value	Never	Current	Past	*p*-Value
Number	432	180	227		432	180	227	
Total energy intake (kcal)	1673 ± 546	1699 ± 565	1766 ± 578	0.121	-	-	-	-
*Macronutrients (g/day)*								
Total protein	64.0 ± 23.2	64.0 ± 23.0	64.9 ± 24.5	0.969	64.8 ± 0.6	64.2 ± 1.0	63.2 ± 0.9	0.358
Vegetal protein	20.4 ± 9.4	19.8 ± 8.5	21.5 ± 8.7	**0.039**	20.7 ± 0.3	19.9 ± 0.4	20.8 ± 0.4	0.201
Animal protein	43.6 ± 19.2	44.2 ± 19.5	43.3 ± 20	0.847	44.1 ± 0.7	44.3 ± 1.1	42.4 ± 1.1	0.356
Carbohydrates	200.7 ± 81.4	200.7 ± 83.1	212.2 ± 83.2	0.154	204.8 ± 2.0	201.7 ± 3.0	204.0 ± 2.8	0.699
Disaccharides	104.1 ± 48.3	108.2 ± 58.4	113.2 ± 56.5	0.232	106.2 ± 1.9	108.3 ± 2.9	109.0 ± 2.7	0.663
Polysaccharides	96.2 ± 52.8	92.1 ± 48.0	98.6 ± 46.8	0.254	98.1 ± 1.7	92.9 ± 2.6	94.5 ± 2.5	0.206
Total fat	63.7 ± 23.6	64.9 ± 24.5	66.4 ± 26.6	0.534	64.9 ± 0.7	65.0 ± 1.1	64.1 ± 1.0	0.778
SFA	23.2 ± 9.7	23.9 ± 11.0	23.7 ± 10.4	0.783	23.7 ± 0.3	23.9 ± 0.5	22.7 ± 0.5	0.152
MUFA	25.8 ± 10.3	26.4 ± 10.2	27.1 ± 12.3	0.581	26.2 ± 0.4	26.4 ± 0.6	26.3 ± 0.5	0.954
PUFA	8.9 ± 4.3	8.9 ± 3.6	9.4 ± 4.3	0.156	9.1 ± 0.2	8.9 ± 0.2	9.0 ± 0.2	0.875
Fiber	16.7 ± 8.7	16.6 ± 8.5	18.5 ± 9.5	**0.016**	17.0 ± 0.3	16.6 ± 0.5	17.9 ± 0.5	0.137
*Micronutrients*								
Cholesterol (mg/day)	275 ± 120	276 ± 133	271 ± 130	0.756	278 ± 5	277 ± 7	264 ± 7	0.231
Calcium (mg/day)	999 ± 490	1040 ± 541	1021 ± 503	0.891	1020 ± 20	1039 ± 30	986 ± 28	0.425
Iron (mg/day)	9.8 ± 3.4	9.8 ± 3.4	10.6 ± 3.6	**0.033**	9.9 ± 0.1	9.9 ± 0.1	10.3 ± 0.1	**0.032**
Vitamin D (μg/day)	2.5 ± 1.8	2.5 ± 1.8	2.5 ± 2.2	0.924	2.5 ± 0.1	2.5 ± 0.1	2.4 ± 0.1	0.901
*Specific foods (g/day)*								
Dairy	211 ± 190	228 ± 195	211 ± 164	0.725	217 ± 8	227 ± 13	199 ± 12	0.281
Red meat	38 ± 34	37 ± 34	39 ± 44	0.933	38 ± 2	37 ± 3	39 ± 2	0.911
Processed meats	10 ± 13	10 ± 13	9 ± 12	0.302	11 ± 1	10 ± 1	8 ± 1	0.119
Wholegrain	51 ± 56	50 ± 54	55 ± 53	0.133	53 ± 3	50 ± 4	54 ± 4	0.733
Fruits ^a^	284 ± 245	298 ± 267	345 ± 303	**0.040**	289 ± 12	301 ± 18	333 ± 17	0.119
Fruits ^b^	320 ± 262	335 ± 277	391 ± 317	**0.016**	325 ± 13	338 ± 19	378 ± 18	0.058
Fruits ^c^	377 ± 289	396 ± 307	453 ± 339	**0.012**	382 ± 14	398 ± 21	442 ± 19	**0.049**
Vegetables	171 ± 110	171 ± 110	196 ± 148	**0.043**	171 ± 6	171 ± 8	194 ± 8	0.056
Fish ^d^	30 ± 26	28 ± 22	31 ± 29	0.850	30 ± 1	28 ± 2	31 ± 2	0.569
Fish ^e^	36 ± 28	34 ± 24	37 ± 33	0.578	36 ± 1	34 ± 2	37 ± 2	0.521
Dietary scores								
Mediterranean ^f^	3.93 ± 1.48	4.04 ± 1.49	4.08 ± 1.45	0.476	3.94 ± 0.07	4.04 ± 0.11	4.06 ± 0.1	0.602
Mediterranean ^g^	4.61 ± 1.86	4.84 ± 1.93	5.03 ± 1.87	**0.038**	4.63 ± 0.08	4.84 ± 0.13	5.00 ± 0.12	**0.036**
AHEI	33.3 ± 10.3	34.8 ± 9.9	35.0 ± 9.7	0.101	33.4 ± 0.5	34.7 ± 0.7	35.0 ± 0.7	0.104
Dietary patterns score								
Meat and chips	−0.29 ± 1.22	−0.45 ± 0.95	−0.40 ± 1.00	0.378	−0.29 ± 0.05	−0.42 ± 0.08	−0.42 ± 0.07	0.212
Fruits and vegetables	0.38 ± 1.55	0.50 ± 1.48	0.77 ± 1.71	**0.017**	0.43 ± 0.07	0.50 ± 0.10	0.68 ± 0.10	0.129
Fat and sugar	0.01 ± 1.44	−0.05 ± 1.33	0.09 ± 1.38	0.484	0.10 ± 0.05	−0.04 ± 0.08	−0.07 ± 0.07	0.146

^a^, fresh fruit only; ^b^, fresh fruit + fresh juice; ^c^, any fruit and fruit juice; ^d^, fish, excluding fried; ^e^, any fish; ^f^, according to Trichopoulou et al. [23]; ^g^, according to Vormund et al. [24]; AHEI, alternative healthy eating index; MUFA, monounsaturated fatty acids; PUFA; polyunsaturated fatty acids; SFA; saturated fatty acids. Results are expressed as average ± standard deviation for bivariate comparisons and as multivariable-adjusted average ± standard error for multivariable comparisons. For dietary patterns, negative scores indicate low adherence to the dietary pattern, whereas positive scores indicate high adherence. Between-group comparisons performed using Kruskal-Wallis test (bivariate) and analysis of variance (multivariable). Multivariable comparisons were performed adjusting for total energy intake (continuous), age (3 categories), education (3 categories), BMI categories (normal, overweight, obese), sedentary level (yes/no) and diabetes (yes/no). Statistically significant p-values are highlighted in bold.

**Table 3 nutrients-11-01930-t003:** Compliance with dietary recommendations and dietary scores according to menopausal hormone therapy.

	Bivariate Analysis				Multivariable Analysis		
	Never	Current	Past	*p*-Value	Never	Current	Past
*Guidelines*							
Fruits ≥ 2/day	209 (48.4)	102 (56.7)	136 (59.9)	**0.011**	1 (ref.)	1.36 (0.93–1.97)	1.35 (0.93–1.94)
Vegetables ≥ 3/day	39 (9.0)	13 (7.2)	24 (10.6)	0.504	1 (ref.)	0.74 (0.37–1.49)	1.06 (0.57–1.98)
Meat ≤ 5/week	296 (68.5)	128 (71.1)	155 (68.3)	0.788	1 (ref.)	1.06 (0.71–1.59)	0.92 (0.62–1.37)
Fish ≥ 1/week ^a^	293 (67.8)	120 (66.7)	161 (70.9)	0.611	1 (ref.)	1.02 (0.70–1.50)	1.28 (0.88–1.88)
Fish ≥ 1/week ^b^	196 (45.4)	80 (44.4)	101 (44.5)	0.966	1 (ref.)	0.99 (0.69–1.43)	1.04 (0.73–1.49)
Dairy ≥ 3/day	40 (9.3)	21 (11.7)	22 (9.7)	0.657	1 (ref.)	1.14 (0.63–2.06)	0.81 (0.44–1.49)
At least three guidelines ^a^	129 (29.9)	64 (35.6)	92 (40.5)	**0.020**	1 (ref.)	1.24 (0.84–1.82)	1.40 (0.96–2.03)
At least three guidelines ^b^	97 (22.5)	48 (26.7)	69 (30.4)	0.078	1 (ref.)	1.20 (0.79–1.84)	1.38 (0.92–2.07)
*Dietary scores*							
Mediterranean ^c^	59 (13.7)	29 (16.1)	38 (16.7)	0.516	1 (ref.)	1.20 (0.73–1.96)	1.38 (0.84–2.25)
Mediterranean ^d^	76 (17.6)	38 (21.1)	51 (22.5)	0.281	1 (ref.)	1.29 (0.80–2.08)	1.42 (0.89–2.26)
AHEI	88 (20.7)	47 (26.1)	59 (26.6)	0.157	1 (ref.)	1.32 (0.85–2.03)	1.45 (0.94–2.24)
*Dietary patterns*							
Meat and chips	115 (27.3)	37 (21.1)	52 (23.6)	0.245	1 (ref.)	0.75 (0.47–1.18)	0.81 (0.52–1.27)
Fruits and vegetables	93 (22.1)	43 (24.6)	68 (30.9)	**0.049**	1 (ref.)	0.99 (0.61–1.60)	1.23 (0.79–1.92)
Fat and sugar	115 (27.3)	36 (20.6)	53 (24.1)	0.209	1 (ref.)	**0.48 (0.28–0.82)**	**0.47 (0.28–0.78)**

^a^, all types of fish; ^b^, excluding fried fish; ^c^, according to Trichopoulou et al. [23]; ^d^, according to Vormund et al. [24]; AHEI, alternative healthy eating index. Results are expressed as a number (column percentage) for bivariate comparisons and as multivariable-adjusted odds ratio and (95% confidence interval) for multivariable comparisons. Between-group comparisons were performed using chi-square (bivariate) and logistic regression (multivariable). Multivariable comparisons were performed adjusting for total energy intake (continuous), age (3 categories), education (3 categories), BMI categories (normal, overweight, obese), sedentary level (yes/no) and diabetes (yes/no). For dietary patterns, the results of the multivariate comparisons are expressed as prevalence rate ratios and (95% confidence interval) of being in the last quartile relative to the other three. Significant (*p* < 0.05) odds-ratios are indicated in bold. Statistically significant p-values are highlighted in bold.

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
