# Peer review of "The Metabolic Benefits of Menopausal Hormone Therapy Are Not Mediated by Improved Nutritional Habits. The OsteoLaus Cohort"

_nutrients, 2019, doi:10.3390/nu11081930_

Round 1

Reviewer 1 Report

This paper is a cross-selectional study retrieved from baseline assessment of the Osteo-Laus Study that is a sub study of the Colaus/PsycoLaus study, investigating the determinants of cardiovascular disease using a population-based sample of post-menopausal women.

In this manuscript the authors studied a sample of 839 post-menopausal women aged 50-80 years subsequently divided in 3 groups: current, past and never users of menopausal hormone therapy (MHT).  The aim of study is to assess whether total energy intake and/or diet quality differ in this cohort of post-menopausal women classified as current, past or never MHT users. Overall it is a nice and interesting paper, the introduction is well done. The material and methods are well structured and described. Also, the sample size is large enough (n=839) to obtain a good level of statistical power.  The detection of dietary data is well conducted. The statistical analysis is well done.  Finally, the discussion is well conducted with mention of limitations and strengths of the study.

Reviewer 2 Report

The authors investigated the effect of metabolic benefits of menopausal hormone therapy (MHT), particularly aiming at testing total energy intake and diet quality as potential mediators of MHT benefits. The study was transversal with 839 participants with complete dietary data selected within a larger cohort study (OsteoLaus study). Women were classified based on their MHT use:“never users” (n=432), current users (n=180), or past users (n=227). They report that the association between MHT use and eating habits are not meaningful, and that other mediators should explain the previously observed improvement in body composition in current MHT users. The question is interesting. The methods are adequate and the paper is very clear.

Major comment

- I don’t agree with the concept of reducing the physical activity behaviors assessed from the physical activity frequency questionnaire to a single yes/no answer to sedentarity, in particular in a study examining metabolic benefits. However, I suppose that you have tested other ways of introducing those variables and that this did not improve the analyses and the understanding of the questions. Table 1 would be more complete if additional data from the PAFQ were provided. Also, line 135, it would be more correct to write sedentary status.

Minor comment

- Although not being native speaker of English and subsequently with potential incomplete understanding, I feel that the sentence in the introduction line 42-43 seems to suggest that reference #6 is an interventional study. I don’t think this is the case.